# Reducing Anxiety and Social Stress in Primary Education: A Breath-Focused Heart Rate Variability Biofeedback Intervention

**DOI:** 10.3390/ijerph191610181

**Published:** 2022-08-17

**Authors:** Ainara Aranberri-Ruiz, Aitor Aritzeta, Amaiur Olarza, Goretti Soroa, Rosa Mindeguia

**Affiliations:** 1Department of Basic Psychological Process and Development, University of the Basque Country (UPV/EHU), 20018 San Sebastian, Gipuzkoa, Spain; 2Department of Clinical and Health Psychology and Research Methodology, University of the Basque Country (UPV/EHU), 20018 San Sebastian, Gipuzkoa, Spain

**Keywords:** stress, anxiety, primary school, breathing, heart rate variability

## Abstract

Primary school students suffer from high levels of anxiety and stress. Having emotional regulation abilities can help them to manage challenging emotional situations. Conscious and slow breathing is a physiological, emotional regulation strategy that is feasible for primary school students to learn. Following Polyvagal Theory and PMER Theory, this research presents the results of a breath-focused heart rate variability biofeedback intervention. The intervention aimed to reduce anxiety and physiological and social stress in primary school children. A total of 585 students (46.4% girls and 53.6% boys) from the same public school, aged between 7 and 12 years (*M* = 8.51; *SD* = 1.26), participated in this study. To assess the impact of training, a mixed design was used with two groups (Treatment and Control groups), two evaluation phases (Pretest and Post-test), and three educational cycles (first, second and third cycles). To examine heart rate variability, emWave software was used and anxiety and social stress were measured by the BASC II test. The results showed that after the intervention, the students learned to breathe consciously. Moreover, they reduced their levels of anxiety (*M*(*SD*)*_pretest_* = 12.81(2.22) vs. *M*(*SD*)*_posttest_* = 13.70(1.98)) and stress (*M*(*SD*)*_pretest_* = 12.20(1.68) vs. *M*(*SD*)*_posttest_* = 12.90(1.44)). The work also discusses the limitations and benefits of this type of intervention in primary schools.

## 1. Introduction

Different studies show that primary school students suffer from significant levels of anxiety and stress [1,2,3]. At this age, stress and anxiety have a very negative impact on physical, psychological, and social problems [4,5], negatively influencing the learning process [6]. They can be considered a predictor of academic failure [7].

Stress refers to the response sparked in the body when the perceived demand exceeds the subject’s own resources or threatens their welfare [8,9]. Everyday stress is defined as frustrating or irritating demands that occur in daily interaction [10]. Within the student population, everyday stressors are grouped into three areas: health, school, and family [11,12]. Regarding the school area: marks, extracurricular activities, academic demands, and learning difficulties create great stress in children [13,14]. Stress is considered an important factor for the appearance of anxiety [15,16], and everyday stressors have been identified as important risk factors for childhood anxiety disorders [17,18,19]. Anxiety is defined as an emotional state characterized by a feeling of uncertainty and malaise caused by a circumstance considered dangerous due to the threat and risk with which it is perceived [7].

The educational system is responsible for ensuring the psychosocial development of students and for training competent citizens [20]. Primary education is fundamental for the teaching–learning process of socio-emotional skills in childhood, such as the regulation of anxiety and social stress levels [21]. Numerous interventions on emotional regulation have been developed in primary education to help young children to learn emotional competence on the path toward effective self-regulation, enhance a positive sense of self and improve their well-being [22].

The emotional regulation program developed in the present study aims to reduce anxiety and social stress in primary school students through a breathing pattern (approximately six pairs of breaths per minute) learned using a biofeedback-based program. Breathing practice is a physiological regulation strategy useful in stressful situations [23]. In such situations, the activation of the parasympathetic nervous system (through the ventral vagus nerve’s nucleus ambiguus) reduces stress [9,24]. Breathing 6 times per minute activates the nucleus ambiguus [25,26], generating a less stressful emotional reaction and influencing the course of the emotional experience [24].

Within the school context, we have only found two interventions similar to the one we use in the present study [27,28]. As in our study, both used the HeartMath EmWave computer application [29] for teaching to breathe consciously by biofeedback of heart rate variability (HRV). However, the small size of the samples and the lack of variability in school years (all were from the same school year) seriously affected the statistical power of their results. Thus, the purpose of this investigation was to conduct and assess an intervention based on learning conscious breathing. Moreover, we expect that the intervention will have a positive impact on reducing anxiety and stress levels in primary school students (between 7 and 12 years).

### 1.1. Polyvagal Theory

According to the Polyvagal Theory [9], emotional reactions are activated by the automatic and unconscious perceptions of the autonomous nervous system of perceived environmental risk. Sensory information from the environment and information about the state of the organism are perceived by the vagus nerve. Depending on whether or not the risk is detected, the vagus nerve will respond. If the vagus nerve detects a safe environment, the parasympathetic activation of the ventral vagus nerve’s nucleus ambiguus will occur, whose fibers connect directly with the sinoatrial node of the heart, slowing down the body’s heart rate (HR).

When risk is perceived, the heart’s parasympathetic influence is deactivated and the sympathoadrenal system is activated. Sympathetic influence on HR is measured by the release of epinephrine and norepinephrine [30], and beta-adrenergic receptors are activated when these hormones are released, leading to the phosphorylation of membrane proteins with cAMP [31]. Thus, in the absence of influence of the ventral vagus on the sinoatrial node, and as a result of activation of the sympathoadrenal system, the HR increases. Although the hematoencephalic barrier keeps the epinephrine from acting on cognitive functions [32], beta-adrenergic receptors in the vagus nerve recapture the norepinephrine in the brain [33,34,35,36], more solidly upholding the experience of stress.

HR is associated with heart rate variability (HRV) (HRV refers to the time interval between heartbeat and heartbeat, which is considered a biomarker of stress) [37]. Therefore, in general terms and barring exceptions, HR and HRV have an inverse relationship; the greater the HR, the lower the HRV, and vice-versa, the lower the HR, the greater the HRV [37,38,39,40,41]. By controlling breathing itself, healthy subjects can modify their own HRV; by breathing slowly and steadily, the ventral vagus’ parasympathetic influence is activated on the sinoatrial node, reducing HR and increasing HRV [42,43,44]. Thereby, by means of physiological self-regulation, emotional regulation facilitates the reduction of anxiety and stress levels [45,46].

In this regard, biofeedback is a method that provides instantaneous information on the variations that occur in some physiological functions [47]. In HRV biofeedback interventions focused on breathing, subjects are taught to breathe at an approximate frequency of six breaths per minute [26]. Thus, they learn to breathe in such a way that their own HRV is spaced out [48,49,50,51]. HRV biofeedback programs based on breathing have proven to be effective in reducing stress and anxiety [26,52,53,54,55].

### 1.2. Theory of the Process Model of Emotion Regulation

Emotional experience is a dynamic process well explained by the Theory of the Process Model of Emotion Regulation (PMER) [56]. This theory is based on the *perception-valuation-action* framework (hereinafter, PVA) by Ochsner and Gross [57], the PVA sequence inherent to the emotional experience is activated with *perception* (P). In this research, we consider that the P is activated through neuroception, based on the hypotheses set forth for the Polyvagal Theory. In this stage of initial perception, sensory inputs are encoded for *valuation* (V). Valuations are conducted with a superimposed set of brain systems that interact and calculate the goodness or badness of the perceptive inputs. There are three different kinds of valuations located in the ongoing cognitive processing, from the most basic to the most elaborate: basic or core valuations are valuations that represent relatively direct associations between perceptions and basic physiological and behavioral responses; at an intermediate level, contextual valuations assess relations between stimuli and responses; and, finally, conceptual valuations represent considerations of stimuli that are abstract and can often be put into words. At any valuation level, the *Action* (A) is activated. This activation can be mental (for example, a memory, a mental image, a thought, etc.) or physical (such as increased heart rate, segregation of certain hormones, etc.). The actions generated have consequences on the surrounding reality in the world (*W*). The PVA processing cycle upholds the emotional reaction. It should be mentioned that multiple PVA cycles operate each time simultaneously. Each cycle maintains its PVA dynamic, and during the time that the emotional experience lasts.

The PMER Theory states that the emotional experience results from the process of the interaction that occurs between emotional reaction and emotional regulation. As mentioned earlier, the emotional reaction is automatic and occurs through the vagus nerve. However, most emotional regulation processes, are conscious and refer to the efforts that we as subjects make to change our own emotions [58]. Emotional regulation consists of three phases: *identification*, *selection*, and *implementation*. Identification is the phase in which the subject experiments with the emotional reaction and decides whether to modify or not this reaction. Regarding the PVA cycle, the perception phase is for detecting the experience of the emotion; the valuation phase is to assess whether the emotional reaction is sufficiently positive or negative to activate regulation, and regulation itself occurs during the action phase. Thus, activation of the objective of the regulation leads to the second phase, which is the selection. The first phase of selection is the perception of different regulation alternatives. These strategies are evaluated in the valuation phase, depending on cognitive ability [59], physiological reality [60], and the type and intensity of the emotional reaction [61,62]. The action phase bears on the application of a determined emotional regulation strategy because it is a change in the subject’s inner world (W). Finally, the implementation stage begins when, after selecting a determined strategy, it is carried out by means of different tactics in the determined situation. 

It is important to consider that emotional experience and emotional reactions have lesser regulation and greater intensity during childhood [63,64] as the cerebral zones, in which cognitive capacities are located, which provide a better understanding of the emotional experience and better emotional regulation, are still developing [65,66,67,68,69]. It should be noted that in the PVA cycle [57], the valuation made (core, contextual, and conceptual) depend on the subject’s cognitive capacities. Thus, given that these capacities are under development, childhood is considered a suitable time to learn emotional regulation [70]. 

In summary, when we perceive risky situations, the sympathoadrenal system automatically activates, generating stress experiences. Based on the PVA cycle [57], PMER Model [56], and the Polyvagal Theory [9], the following occurs: we perceive (P) the automatic reaction generated through neuroception; we valuate (V) with the cognitive resources available at that point in development, the goodness or badness of reality; different actions (A) are sparked in the body, from HR activation to the decision to regulate this state. Therefore, in terms of emotional regulation (based on the PMER model) [56], children that have been trained in the emotional self-regulation process could identify their state, *select* to breathe slowly and steadily, the implement steady breathing at approximately six breaths per minute, which would reduce the impact of the stressful situation on the body by activating the ventral vagus, slowing HR and increasing HRV, consequently reducing stress and anxiety and generating a new situation (W) wherein different PVA cycles would occur [71]. In this sense, interventions with breath-focused HRV biofeedback are suitable since they help to reduce HRV.

For this reason, and given that in the educational realm the importance of well-being is acknowledged [72,73,74] and that interventions in emotional regulation are recommended in primary school to improve well-being [75,76,77,78], the objective of this research was to develop a school program where students will learn to breathe steadily approximately six breaths per minute. This training process will be monitored by a biofeedback technique where HRV changes will be informed to participants. As it was argued, we expect that our program will also positively impact students’ emotional regulation by reducing their stress and anxiety levels.

## 2. Materials and Methods

### 2.1. Participants

In total, 585 students (46.4% girls and 53.6% boys) participated in this study. All the children were from the same public school, aged 7 through 12 (*M*(*SD*) = 8.51 (1.26). The sample was divided by the Primary School cycles: 21.4% (*n* = 125) belonged to the first cycle (2nd course, about 7 years old), 64.6% (*n* = 378) to the second cycle (3rd and 4th courses, 8–9 years old), and the remaining 14% (*n* = 82) were third-cycle students (5th and 6th courses, 10–11 years old). The number of participants in each school year corresponds to the natural sample size of each course. In this sense, the sample was incidental and no randomization was possible. As the groups were already formed, the only possible decision for the research team was the assignation of those natural groups to control and treatment conditions. This decision also involved the teachers, the Head of Therapeutic Pedagogy, and the School’s Management Team. Thus, in the first cycle, there were 83 participants belonging to the treatment group and 42 in the control group; in the second cycle, 257 participants in the treatment group and 121 participants in the control group; and lastly, in the third cycle, 49 participants in the treatment group and 33 in the control group.

The participation was voluntary and consented to by the school board, parents, and guardians. The study had a favorable report from the ethics committee for research with humans, their samples, and their data (CEISH/269 1-2-3-4-/2014) from the University of the Basque Country/Euskal Herriko Unibertsitatea (UPV/EHU) with the DSI file INA0079. Throughout the entire research, ethical aspects required for research with humans were scrupulously followed (personal data protection, informed consent, right to information, confidentiality guarantees, non-discrimination, no cost, and the possibility to leave the study during any of its phases).

### 2.2. Design

The biofeedback treatment consisted of 5 individual sessions. To assess the impact of training on the reduction of anxiety and social stress in students, a mixed design was used consisting of two groups (treatment y control), two evaluation phases (pretest and post-test), and three educational cycles (first, second, and third cycles). Group and educational Cycles are the inter-participant measurement factors, and the evaluation factor is repeated or intra-participant measurement.

The dependent variables were the high HRV number and the anxiety and stress levels of the participants measured with the BASC-S2 test [79].

The intervention began after the Christmas holidays. This choice was determined by the School’s Management Team (together with the research team) based on the consideration that the students showed less tension after the holidays.

### 2.3. Instruments and Materials

The instruments and materials used were selected based on their suitability for measuring stress and anxiety. Considering the three-dimensional nature of stress (biological, psychological, and social) and following the recommendation to use different kinds of measurement [80], we used HRV as a biomarker for stress and BASC-S2 as a self-reporting measure both for stress and for anxiety. In order to stand in for possible deficiencies caused by the characteristics inherent to self-reporting measurements, such as the influence of possible biases of respondents (memory and social desirability biases lead to underestimating and overestimating) [81] and the limited variance proportion that is explained by not considering the biological dimension of stress [82]. We used HRV as a biomarker for stress because of the benefits provided [39,40,83].

To record HRV values, we used ad hoc record sheets where tutors register the HRV numbers shown by students immediately after the end of each session (final result). However, it should be mentioned that students were able to see their progress while they were connected to the emwave system (biofeedback). Each participant had 5 HRV measures. The first one, when the baseline that is taken in the pretest phase, the second, third and fourth that are taken during the intervention, and finally, the last one, once the intervention is finished, in the post-test phase.

The dimensions of anxiety and social stress were measured before and after training with BASC-S2 (Self-Report) “Behavior Assessment System for Children” [79], adaptation to Basque [84]. This tool provides information on clinical and adaptive scales. The social stress scale (SS, *α* = 0.71) is defined as the stress level experienced by girls and boys in their interactions with others. The anxiety scale (AN, *α* = 0.83) is defined as the state characterized by feelings of worry, nervousness, and fear. The responses to the items are dichotomous (True = 1/False = 2). It should be mentioned that a higher score in the dimension means a lower presence of the dimension, and vice-versa. The BASC-S2 was completed by all students one week before (pretest phase) and one week after the intervention (post-test phase).

The computer software used for participating students to learn to breathe an average of six breaths per minute was EmWave 2021 Pro. Version(Copyright HeartMath, Inc. Quantum Intech, Boulder Creek, CA, USA). This software’s efficacy in learning breathing patterns to increase HRV has been proven by different studies [27,48,49,85]. The software requires digital equipment, a USB, and a non-invasive auditory sensor connected to the earlobe used to detect the HRV in real-time. The HRV numbers are divided into three colored groups: low (red), medium (blue), and high (green). It gives a total HRV of 100 points distributed on low, medium, and high HRV in real-time. Thus, when breathing slowly and steadily, the green scores increase, while when breathing hurriedly, the red numbers go up. Points that are not distributed to the green or red group are assigned to the blue group. Upon completion of each connection, the software provides average HRV information, divided into the aforementioned colors (red is low, blue is medium, green is high) for the time interval used. Data from green (for high HRV) were processed for statistical analysis.

The applications from the HeartMath emWave program used to learn to breathe slowly and steadily were: Balloon Game and Coherence Coach, where students learn to breathe at approximately six breaths per minute in a fun way.

The individual sessions were held in a relaxed location at the school that was properly outfitted. In this place, there were two chairs (one for the tutor and another for each student), a desk with a computer with the HeartMath emWave software installed on it, a USB, and an ear sensor.

### 2.4. Procedure

The program carried out has two implementation phases. On one hand, the pre-intervention phase is an 8-h phase to train teachers. In this phase, teachers receive the training they need from research team staff to learn how to use the Coherence Coach and Balloon Game programs, so they can conduct the intervention with students and collect data on each student on the record sheets. 

On the other hand, and only in relation to the treatment group, the intervention phase consisted of conducting one session per week for 5 consecutive weeks. Weekly sessions were individual and always occurred in the same space provided to this end. Each session lasted approximately 15 min. This way, under the guidance of a previously trained tutor, each student learned to breathe approximately 6 breaths per minute with the aforementioned applications throughout 5 sessions. It should be mentioned that, in order to generalize lessons learned, in the third session, each student was provided with a “target” image (from the program), laminated and sized 6 × 4 cm. They carried this in their school pouch for use when teachers recommended it and when they felt nervous. This image was to help them to generalize breathing slowly and steadily (approximately 6 breaths per minute).

## 3. Results

Firstly, a normalcy analysis was conducted with the Kolmogorov–Smirnov test. The results showed that there was normalcy in data distribution.

Results on the efficacy of biofeedback training show that, from measuring HRV at the beginning of the training until the final evaluation, students in the treatment group learned to breathe slowly and steadily (*F*(1.97) = 176.26, *p* < 0.001; $\eta_{p}^{2}\;$= 0.372). This learning occurred in a significant fashion in all educational cycles (*F*(2.297) = 11.10, *p <* 0.001,$\;\eta_{p}^{2}=$ 0.070). And as observed in the interaction *F*(2.297) = 21.05; *p <* 0.001; $\eta_{p}^{2}=0.124$ students from the second cycle (HRV session 1: *M*(*SD*) = 23.78 (29.75); HRV session 5: *M*(*SD*) = 94.99 (60.56)) were the ones who showed the greatest improvement from the first measurement until the last measurement of HRV obtained after the intervention (see Table 1 for the results). 

Regarding BASC-S2, dichotomous answers were encoded thus: True/False = 1 and No/False = 2. In this test, high dimensions scores are associated with lower dimensions levels and vice-versa.

To examine the impact of training on HRV biofeedback for stress and anxiety reduction in children, with the BASC-S2, we analyzed the anxiety dimension (AN) and the social stress dimension (SS). The results were analyzed with ANOVA tests for blended designs, 2 (group: control, treatment), ×2 (Evaluation pre- and post-training), ×3 (educational cycle: first, second, and third). Group and educational cycle were variables with independent measurements, and evaluation had repeated measurements. Post hoc comparisons were conducted with the Bonferroni test and comparisons with pairs of student *t.* Finally, it should be mentioned that to interpret low, medium, and large effect sizes, we followed the Kelley and Preacher criteria [86].

### 3.1. Anxiety

The anxiety score in the post-test was higher than in the pretest (13.24 vs. 12.88) (*F*(1.47) = 8.42, *p<* 0.05, $\eta_{p}^{2}$= 0.018). The group factor was significant (*F*(1.47) = 8.37, *p<* 0.05, $\eta_{p}^{2}\;$= 0.018). And the interaction evaluation × group was significant (*F*(1.47) = 18.80, *p*< 0.001, $\eta_{p}^{2}\;$= 0.039). In the initial evaluation, the control group (*M*(*SD*) = 12.95 (2.16)) and the treatment group (*M*(*SD*) = 12.81 (2.22)) obtained similar anxiety scores without statistically significant differences (*t*(503) = −0.73, *p* > 0.05). After the biofeedback intervention, the control group (*M*(*SD*) = 12.79 (2.45)) obtained a lower score than the treatment group (*M*(*SD*) = 13.70(1.98)). This difference was statistically significant with a moderate effect size (*t*(488) = 4.58, *p* < 0.001, *d* = 0.42). In turn, the group that received the intervention reduced their anxiety level from the pretest measurement (*M*(*SD*) = 12.81 (2.22)) until the post-test measurement (*M*(*SD*) = 13.70(1.98), *t*(231) = −6.58, *p* < 0.001, *d* = −0.42), while the control group worsened their anxiety level between scores obtained before the intervention (*M*(*SD*) = 12.95 (2.16)) and after the intervention (*M*(*SD*) = 12.79 (2.45), *t*(242) = 0.837, *p* > 0.05, *d =* 0.07). 

Moreover, the educational cycle factor was significant (*F*(2.469) = 11.86, *p* < 0.001, $\eta_{p\;}^{2}$= 0.048). Post-hoc comparisons with the Bonferroni test showed that there was a significant increase in anxiety from the first cycle (*M*(*SD*) = 13.77(0.18)) through the second cycle (*M*(*SD*) = 12.97(0.10)), and from the first cycle through the third cycle (*M*(*SD*) = 12.46(0.22)), with no significant difference between the second and the third cycle. Although all educational cycles that received treatment significantly reduced their anxiety levels (Cycle 1: *t*(47) = −3.55, *p* < 0.001, *d =* −0.54; Cycle 2: *t*(152) = −4.78, *p* < 0.001, *d =* −0.37; Cycle 3: *t*(30) = −2.98, *p* < 0.05, *d =* −0.60). 

No significant interactions were observed in evaluation × educational cycle interactions, nor in evaluation × educational cycle × group interactions. See the means and standard deviations in Table 2.

### 3.2. Social Stress

The score on social stress in the post-test phase was lower than in the pretest one (12.21 vs. 12.35) (*F*(1.461) = 4.55, *p <* 0.05, $\eta_{p}^{2}\;$= 0.010). Although the group factor was not significant (*F*(1.46) = 10.07, *p* > 0.05, $\eta_{p}^{2}\;$= 0.005), the evaluation × group interaction was statistically significant (*F*(1.46) = 36.22, *p*< 0.001, $\eta_{p}^{2}\;$= 0.073). In the initial evaluation, the control group (*M*(*DT*) = 12.22(1.67)) and the treatment group (*M*(*SD*) = 12.45 (1.63)) obtained similar social stress scores without statistically significant differences (*t(*503)= −1.55, *p >* 0.05). After the biofeedback intervention, the control group (*M*(*SD*) = 12.23 (2.06)) obtained a lower score than the treatment group (*M*(*SD*) = 12.93 (1.42)). This difference was statistically significant and with a moderate effect size (*t*(480) *=* 4.37, *p <* 0.001, *d =* 0.40). In turn, the group that received the intervention reduced their stress level from the pretest measurement (*M*(*SD*) = 12.20 (1.68)) to the post-test measurement (*M*(*SD*) = 12.90 (1.44), *t*(230) = −5.96, *p* < 0.001, *d* = −0.45)). In the control group, just like what happened with the anxiety dimension, social stress increased from the measurement before the intervention (*M*(*SD*) = 12.49 (1.64)) until after the intervention. This worsening was statistically significant (*M*(*SD*) = 12.21 (2.07), *t*(235) = 2.13, *p < 0.*05, *d =* 0.151).

The educational cycle factor was not significant (*F*(2.461) = 0.50, *p* > 0.05, $\eta_{p}^{2}\;$= 0.002, and post hoc comparisons with the Bonferroni test showed no significant changes in the social stress levels between the three cycles. The interaction between evaluation × educational cycle × group was significant (*F*(2.461) = 3.372, *p* < 0.05, $\eta_{p}^{2}\;$= 0.014). Regarding the treatment group, we can observe how all cycles show less levels of social stress, with the particularity that, in the third cycle, an improvement with a large effect size occurs (Cycle 1: *t*(47) = −2.756, *p* < 0.005, *d = −*0.52; Cycle 2: *t*(152) = *−*4.096, *p* < 0.001, *d*= *−*0.341; Cycle 3: *t*(29) = −3.641, *p* < 0.001, *d = −*0.840). However, in the control group, in all cycles, the social stress level increased. As such, we observed that the intervention, as it happened in the anxiety dimension, also managed to invert increased social stress that was naturally occurring in all primary school cycles. All these results can be seen in Table 2.

## 4. Conclusions

The purpose of this research was to design and implement an HRV biofeedback intervention based on breathing to teach primary school students to reduce their own HRV by practicing slow and steady breathing (six breaths per minute, approximately) and, consequently, reducing their anxiety and social stress.

The research objective was based, on the one hand, on the PMER Theory [56] regarding emotional regulation as a resource to reduce the emotional impact of stressful experiences in the PVA valuation cycle [57]. On the other hand, in the Polyvagal Theory [9], where emotional reactions related to stress are automatically generated by deactivating the ventral vagus nerve and activating the sympathoadrenal system. HRV is an indicator of stress, and we assume that practicing voluntary calm breathing is a resource to regulate the emotional experience related to stress and anxiety.

The results showed that, after the intervention, primary school students who participated in the program managed to reduce their previous physiological stress (HRV), social stress, and anxiety (BASC-S2) levels in a significant way. The reduction in anxiety and social stress levels is also statistically significant when compared with the results shown by the control group during the post-test in both dimensions. In the control group, we observed that the simple passing of time between the pretest and the post-test measurement leads to an increment in anxiety and social stress scores. Thus, we can conclude that the intervention not only improved anxiety and physiological and social stress numbers but also inverted the trend toward increased anxiety and social stress that occurred with the analyzed primary school students.

In other words, interpreting these results from the Polyvagal Theory [9] and the PVA valuation process [57], after the intervention, when participant students are in a stressful situation (W-PVA cycle) and the sympathoadrenal system automatically activates (P-PVA cycle), they will valuate (V) the goodness and badness of the experience, leading to different actions (A) such as increased HR. Consequently, based on the PMER model [56], after identifying their stressful state, the student shall select the breathing strategy and implement breathing six times per minute. Therefore, this will affect their environment (W), since, inside their body, the ventral vagus will begin to activate, along with its parasympathetic influence on the sinoatrial node, reducing HR and increasing HRV, which means a reduced impact for stress.

Some interesting results also arise when comparing cycles (groups of students of different years). Students from the second cycle (between 9–10 years old) showed the greatest improvement in HRV. Regarding anxiety, a significant increase in anxiety from the first cycle through the second cycle and from the first cycle through the third cycle was also observed. Finally, regarding the social stress dimension, all cycles show lower levels of social stress in the post-intervention measure, and in the third cycle, an improvement with a large effect size occurred. From an exploratory point of view, we can conclude that the intervention conducted had a bigger impact on children 8 years old and above. These differences between cycles may be justified by maturity factors. Specifically, heeding to the neuro-anatomical cerebral zones for emotional self-regulation (for example, the anterior ventral cingulate cortex and the prefrontal ventromedial cortex), and being aware of their developmental process between childhood and adolescence, we can infer that this lower neuro-anatomical maturity progress at 7 years makes it difficult, for example, to acquire the emotional self-regulation skill measured in this case through increased high HRV.

Having taken into account the results obtained and understanding the consequent dynamic, it can be said that the HRV biofeedback intervention conducted with the HeartMath Enwave computer program was an effective strategy to modify how children breathe, leading to reduced HRV.

Prior studies conducted with primary school students also show the efficacy of these interventions conducted with the same technology. For example, Bothe et al. [27] conducted an HRV biofeedback program for 4 months with individual daily sessions lasting 10 min, with 13 students in the 3rd year of primary school. They used “The revised children’s manifest anxiety scale” (RCMAS) [87]. The measures were analyzed at three different times: before the intervention, immediately after the intervention, and one year after the intervention. The results showed that participant students reduced their anxiety in comparison with their previous measurements and in comparison with the control group after the intervention and one year after the intervention. Cruz [28] also conducted an HRV biofeedback intervention with 14 students in the 4th year of primary school for six weeks with weekly individual sessions lasting 45 min in order to increase student well-being. They used the BASC 3 TRS [88] to measure the teachers’ perception of student’s condition and the “Self Assessment Manikin” (SAM) to measure the self-evaluation of participating students [89]. The results showed that improvements occurred in the treatment group after training.

The scientific literature considers that breathing-focused HRV biofeedback interventions are effective in reducing stress and anxiety in primary school students. From a neuropsychological perspective, emotional regulation strategies used by primary school children change throughout the educational cycle [21], depending on the development of brain areas responsible for executive functions, which are located in the prefrontal cortex, a developing area from early childhood to adolescence [90]. In the last two decades, multiple cognitive–emotional regulation interventions have been developed, based, for example, on theoretical frameworks such as the Social Information Processing theory (SIP) [91], “Tools For Getting Along” [92], or “Making Choices” which have shown very good results [93].

Therefore, based on the results observed in this research, together with the results observed in other interventions, we consider that the development of breathing-focused HRV biofeedback interventions are helpful in reducing anxiety and stress levels in children in primary schools.

### Limitations and Future Proposals

We should mention certain limitations of this study and propose future research lines stemming from our intervention. The main limitation of this study is the configuration of the treatment group, as it was not randomly created. However, in previous studies [85], it has been shown that natural composed groups can also benefit from HRV biofeedback interventions. As in previous interventions [28,48], the natural configuration of the groups was respected, and the corresponding sample size made the analysis possible. In future studies, it would be advisable to use a power analysis to determine the sample size and, additionally, to use randomized controlled trials (RCT). Doing this would increase the power of the investigation and would ensure, to a greater extent, that the only difference between the two groups (treatment and control) is their participation in the intervention.

Regarding the timeline, we should state that, given that the main objective was to examine the impact of our intervention from a cross-sectional perspective in an entire primary school cycle, we did not conduct, for example, the study on the longitudinal impact of the intervention (conducted, for example, with satisfactory results in de Bothe et al.’s [27] intervention) and the possible influence of other variables. Regarding the influence of other realities on HRV and having observed that maternal attachment plays a mediating role in HRV [94], it would be advisable to analyze the influence of mother–father relationships with children on HRV and well-being. Along these same lines, given that the type of parental socialization has an influence on children’s psychosocial adjustment [95], we should analyze its influence on well-being.

Moreover, related to anxiety and social stress in primary school children, different studies have shown that perceived social support, especially from the peer group, is related to an increase in well-being [96] and in a reduction in psychological stress [97]. Therefore, it could be interesting to include the perceived group support as an additional variable to take into account in the study. When also taking into consideration that some school periods are more stressful than others [98,99], it is worth considering in which school year trimester the intervention is implemented. In addition, it would be useful to ask about the socioeconomic characteristics of the children in the socio-demographic questionnaire, given that these can become a stress factor for children [100,101].

In relation to the school type (private or public), it is worth mentioning that the participant centre is a public school. This school belongs to the “Amara Berri” group, whose philosophy includes the consideration of facilitating the development of children in a comprehensive manner [102]. For future research, it would be useful to analyze whether belonging to this group (or schools that are similar to its philosophy) influences the impact of the intervention. Along the same lines, given that the center where it was developed is publicly funded, and given that there were more private than public primary schools in northern Spain [103], it would also be advisable to analyze whether the public-private typology affects the results of the intervention.

With regard to teacher’s characteristics, it has been observed that the stress experienced by teachers has a negative effect on student well-being [104]. It has also been observed that teacher’s well-being influences student’s well-being [105]. In this sense, we also consider that future research should try to include teacher’s characteristics to analyze the possible influence of these variables on the impact of the intervention.

Regarding the multidimensionality of the information and data collection, we also suggest using additional direct biomarkers such as EEG signals to measure the impact of stress through brain waves. Also, the use of hetero-reported data through qualitative observations from teachers using other standardized tools (such as BFRIEF-2, EDAH, Aula Nesplora, etc.) would be interesting.

Finally, we also consider that it is highly important to analyze whether improved well-being has any impact on school performance, school environment, and other realities inherent to the learning process that occur at the educational center.

To end with, and taking the results and limitations into account, it can be mentioned that our biofeedback intervention has shown to be a suitable strategy to reduce anxiety and social stress in primary school. Slow and steady breathing HRV biofeedback-based intervention has shown to be a suitable regulation strategy [49,53] and has an impact on stressful experiences, affecting the PVA valuation cycle [57] inherent to the emotional experience.

## Figures and Tables

**Table 1 ijerph-19-10181-t001:** HRV means (*M*) and standard deviations (*SD*) for the three primary school cycles.

	Cycle 1	Cycle 2	Cycle 3	Total
s	***M (SD*) **	***M (SD*) **	***M (SD*) **	***M (SD*) **
**Session 1**	26.61 (31.93)	23.78 (29.75)	13.16 (19.88)	23.25 (29.55)
**Session 5**	49.77 (51.10)	94.99 (60.56)	76.39 (54.84)	79.52 (60.47)

**Table 2 ijerph-19-10181-t002:** Means (*M*) and standard deviations (*SD*) for the treatment and control group in the pretest and the post-test for the three primary school cycles.

		Cycle 1		Cycle 2		Cycle 3	
		Treatment	Control	Treatment	Control	Treatment	Control
		*M(SD)*	*M(SD)*	*M(SD)*	*M(SD)*	*M(SD)*	*M(SD)*
AN	Pre	13.54 (1.82)	13.78 (1.71)	12.67 (2.18)	12.90 (2.10)	11.94 (2.57)	12.32 (2.71)
	Post	14.46 (1.61)	13.30 (2.46)	13.47 (2.07)	12.83 (2.51)	11.88 (1.93)	13.69 (1.78)
	Total	*M(SD)*	*M(SD)*	*t*	*df*	*p*	*d*
	Pre	12.81 (2.22)	12.95 (2.16)	−0.73	503	>0.05	-
	Post	13.70 (1.98)	12.79 (2.45)	4579	488	<0.001	0.42
SS	Pre	12.04 (1.47)	12.50 (1.38)	12.31 (1.73)	12.49 (1.66)	12.47 (1.81)	11.93 (1.78)
	Post	12.81 (1.49)	12.00 (2.03)	12.85 (1.43)	12.34 (2.07)	11.85 (2.07)	13.27 (1.41)
	Total	*M(SD)*	*M(SD)*	*t*	*df*	*p*	*d*
	Pre	12.45 (1.63)	12.22 (1.67)	−1.55	503	>0.05	-
	Post	12.93 (1.45)	12.23 (2.06)	4.37	480	<0.001	−0.44

AN = anxiety. SS = social stress (BASC-S2).

## Data Availability

Data is available on request from the corresponding author.

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
