# Peer review of "Reducing Anxiety and Social Stress in Primary Education: A Breath-Focused Heart Rate Variability Biofeedback Intervention"

_ijerph, 2022, doi:10.3390/ijerph191610181_

Round 1

Reviewer 1 Report

This study is very practical since it aims (and succeeds) to present and statistically validate a school program for reducing anxiety and stress through learning "more conscious breathing". The proposal is specified in a biofeedback intervention of variability of the heart rate centered on respiration. We highlight very positively in this study: the breadth of the sample, balanced sample according to gender, specific age range, ... and above all, the methodology: mixed design with Experimental and Control Group, two evaluation phases (Pre-Test and Post-Test), and three Educational Cycles (first, second and third cycle) of evaluation of the results of the program The results show the relationship between learning to breathe consciously and the reduction of anxiety and stress levels physiological and social.

Some recommendations for improvement could be the following. The research focuses on students from a single public school in a region of Spain (Basque Country) with a high level of schooling and innovative teaching programs (with outstanding results in the PISA report). It would have been desirable to have a sample of students from other schools, not only public but also private. On the other hand, it would be advisable in the future to compare these results with those of other Spanish regions with worse academic results, to check to what extent the rate of schooling and the level of skills of the students influence their management of stress and anxiety. These two recommendations do not undermine the purpose of the study but rather encourage further research in this area. We believe that this study is publishable in its current structure and contents.

Reviewer 2 Report

Abstract

·       Citations have been included in the abstract, which is unusual. Please check this is acceptable to the journal.

·       I would recommend adding statistics to support the results described in the abstract (e.g., mean change from baseline and p values for each reported outcome).

Introduction

·       Minor point - on page 2 that the population targeted in this study were aged 6-12 years, whereas in the abstract it says participants were aged 7-12 years, please could the authors clarify which age range is correct?

·       Sections 1.1 and 1.2 it is useful that the authors have identified theories which potentially explain how the intervention could reduce stress and anxiety. However, not coming from a neuroscience background, I found these sections difficult to follow. Would it be possible to reduce the length of these sections and to simplify to make the information more readable for non-specialists?

Materials and methods

·       I assume this study was undertaken in Spain. Please could the authors clarify if a ‘public school’ refers to one that is funded by the government or privately by parents (the terms public school and private school mean the same thing in the UK). This would be helpful to understand the socioeconomic profile of participants and generalisability of results to the wider population.

·       Could the author please clarify what is meant by first, second and third cycle and why students were split into these groups?

·       The authors noted that small sample size is a limitation of previous studies. Please can the authors clarify if a power calculation was undertaken to inform the number of participants for the present study? If not, this should be noted as a limitation.

·       The self-selecting nature of the sample could introduce selection bias, compromising representativeness of the sample and the generalisability of the results, which should be noted as a limitation.

·       Given that participants do not appear to have been randomly assigned to study arms, could the authors tabulate characteristics of participants in each intervention and control group, including potentially confounding factors such as socioeconomic status? This would enable the reader to assess whether groups differed in ways other than the intervention. In the discussion, the superiority of RCTs for assessing intervention effectiveness could also be mentioned and suggest as a suitable design for further research on this topic.

·       It is good that subjective and objective measures were used. Could the authors provide information regarding the validity and reliability of these measures and why they were chosen over alternative methods? Alpha values are provided for the BASC but I am not sure what these values refer to.

·       Is it possible for the authors to provide a link to the BASC tool and methods paper, to clarify my understanding of the structure of this assessment, how it is implemented and how the results were obtained?

·       Please could the authors tabulate the HRV results as they have done for the BASC results, to allow the reader to easily compare HRV results between assessment timepoints? I am also not clear which statistical tests were performed on the HRV data, therefore I am struggling to interpret the results. Please could the authors clarify this?

·       Table 1: please could the authors clarify whether a higher score is better (reduced anxiety) or worse (increased anxiety) and add p values to the tables? Also, is it possible to provide overall results across the 3 cycles as appears to have been done in the text?

Reviewer 3 Report

The paper deals with the analysis of the impact of heart rate variability biofeedback intervention focused on primary school students. The results support the conclusion that a statistically significant reduction of physiological stress, social stress and anxiety levels was achieved by the students  participating to the intervention.

There are some aspects peculiar to the school settings that in my opinion should be better explained by the authors in order to assess whether further variables  usually playing a role in the anxiety levels can influence the conclusions. In particular it would be interesting to know when the intervention took place (beginning of the school year, intermediate months, end of the year). Usually anxiety levels tend to reach a maximum in the beginning and final part of the school year.

Moreover it would be beneficial to the reader to know whether  within-group variations for groups with the same teachers are large as compared to between-group variations. A large between-group variation might point to teachers-related issues as cause for an increased anxiety level.

Long-term control of anxiety level in school settings usually requires to deal with the underlying explaining factors (if present), in particular teachers' behaviour and strategies.

Should these data be available it would be useful to discuss them in deeper detail in the manuscript.

Reviewer 4 Report

1.       I have a number of concerns for this manuscript, which I feel can be recified if the authors modify narrative

2.     Introduction: The narrative throughout the introduction lacks nuance and detail specific to the study that was undertaken. The introduction fails to mount a sufficient cogent argument for why another treatment for the anxiety and/or stress is needed and what benefit this new treatment has over existing treatments.

3.     A good deal of the narrative is descriptive in nature, with numerous non sequitur statements.

4.     Areas of interest, stress and anxiety are broadly defined but it appears the authors developed a treatment for social anxiety.

5.     Review of two theories needs to be concise and applied to the study. At present it reads as a description of each theory which is more relevant to a chapter. Readers are sophisticated and will understand the theories. The authors should focus on why the theories are relevant to the study at hand.

6.     Why no hypotheses or apriori power analysis to determine sample size?

7.     Method
Primarily studies of this type (Experimental) should be described as per CONSORT guidelines (http://www.consort-statement.org/). The authors should reframe this section to do so.

8.     No explanation of how participants were organized into experimental and control groups

9.     No explanation for the varying numbers in each cycle. Where there different cohorts for each cycle, if yes why, if not what is the rationale?  

10.  What was the randomization machine.

11.  Given the study was focused on using biofeedback to regulate breathing, was there any preselection criteria? That is, was a selection a medical diagnosis of social stress and/or anxiety present in the sample? If not, this might be a fatal flaw in the results. If no pre-selection criteria were present, why subject students of this age to a treatment when they showed no signs of experiencing social stress and/or anxiety?  

12.  Results

13.  No information on the descriptive scores as to how they compare to standardized norms for each inventory.

14.  Inferential statistics should be presented against set hypotheses (which are not present in the paper). This will help readers relate the statistical outcomes to the objectives of the paper.

15.  Check notation format for the presentation of statistical outcomes. At present they do not meet accepted guidelines.

16.  No explanation was given for changes in social stress and anxiety between cycles. This seems to be contrary the claim of success of the treatment.

17.  Some narrative needed to put into context the claim of large effect sizes based on the measurement scales.

18.  In terms of temporal measurements, how close was the HRV measurement to the biofeedback technique. Moreover, how long did the treatment buffer social stress and anxiety symptoms?

Round 2

Reviewer 4 Report

The Authors have adequately addressed the reviewer's initial critique of their paper. 

Given the authors have identified their study as non-experimental, as is the case, they should avoid using the term experiemental in their manuscript and in tables. Continued use of this term could cause misunderstanding in readers' interpretations. I would suggest either the terms treatment or intervention to replace the term experimental in the manuscript to best reflect the design of the study. 

Author Response

Thank you!

As you suggested, we have avoided using the term experimental and have replace it by the term treatment (group or condition).